# Some Remarks about Entropy of Digital Filtered Signals

**DOI:** 10.3390/e22030365

**Published:** 2020-03-23

**Authors:** Vinícius S. Borges, Erivelton G. Nepomuceno, Carlos A. Duque, Denis N. Butusov

**Affiliations:** 1Control and Modelling Group (GCOM), Department of Electrical Engineering, Federal University of São João del-Rei, São João del-Rei 36307-352, Brazil; viniciussb@ufsj.edu.br; 2Department of Electrical Engineering, Federal University of Juiz de Fora, Juiz de Fora 36036-900, Brazil; carlos.duque@ufjf.edu.br; 3Youth Research Institute, Saint-Petersburg Electrotechnical University “LETI”, Saint Petersburg 197376, Russia; dnbutusov@etu.ru

**Keywords:** theory of information, computer arithmetic, digital filter, shannon entropy

## Abstract

The finite numerical resolution of digital number representation has an impact on the properties of filters. Much effort has been done to develop efficient digital filters investigating the effects in the frequency response. However, it seems that there is less attention to the influence in the entropy by digital filtered signals due to the finite precision. To contribute in such a direction, this manuscript presents some remarks about the entropy of filtered signals. Three types of filters are investigated: Butterworth, Chebyshev, and elliptic. Using a boundary technique, the parameters of the filters are evaluated according to the word length of 16 or 32 bits. It has been shown that filtered signals have their entropy increased even if the filters are linear. A significant positive correlation (*p* < 0.05) was observed between order and Shannon entropy of the filtered signal using the elliptic filter. Comparing to signal-to-noise ratio, entropy seems more efficient at detecting the increasing of noise in a filtered signal. Such knowledge can be used as an additional condition for designing digital filters.

## 1. Introduction

Digital filters are discrete-time maps that perform mathematical operations on a sampled signal [1]. Frequency response is usually applied to characterize filters [2,3]. Two main classes of digital filters are generally used. When an impulse response is not zero for a finite number of samples, then we have the finite impulse response (FIR) filters. In the case where the impulse response produces an infinite number of non-zero samples, then we have the infinite impulse response (IIR) [4,5]. The great performance of digital filters is believed to be one of the reasons explaining the popularity of DSP devices [6].

The process of digital filtering is extensively used in many applications in communications, signal processing, electrical and biomedical engineering, and control [7,8,9,10,11,12,13,14,15]; for example, coding and compression, signal augmentation, denoising, amplitude and frequency demodulation, analog-to-digital conversions, shape detection, and extraction [16,17,18,19,20,21,22,23,24,25]. For some applications, nonlinearity is tailored to a specific purpose [26]. Recently, the authors of [27] designed a digital sigma–delta truncated infinite impulse response filter, which furnishes adequate rejection with a digital-to-analog converter of no more than 8 bits. The application in [27] is related to human body communication, which for many researchers is a promising research topic as it plays an important role in wireless body area networks because of its low power and hardware cost. In this area, it seems that digital filters of medium to low word length has again attracted the attention of researchers.

When digital filters are employed under fixed-point arithmetic platforms, e.g., microcontrollers, DSP, and FPGA, or with very demanding performance specifications, the importance of filter coefficient accuracy increases, because the signal may be distorted [28,29,30]. Thus, a common goal in the finite precision analysis is to choose a word length such that the digital system presents sufficiently accurate realization. This design should consider the complexity and cost of hardware and software [31].

In digital signal processing, the issues of finite word length are some of the most significant components when the discrete poles are very close to the unit circle. Mullis and Roberts [32] and Hwang [33] have demonstrated that the influence of quantization errors on the digital filter performance depends on the filter implementation. In addition, Rader and Gold [34] have shown that for a given filter implementation it is possible that small errors in the denominator or numerator coefficients may cause large pole or zero offset. Moreover, Goodall and Donoghue [35] and Jones et al. [36] have observed a significant sensitivity of coefficient word lengths. This fact relates to the inability of computers to represent the infinite nature of real sets [37]. The influence of computer limitations opens a new perspective for computer environment simulation. For example, Nepomuceno [38] presents a theorem that identifies the reliability of calculations performed at fixed points; in [39,40], a technique has been developed to decline a simulation if a mandatory accuracy is greater than the lower bound error, growing numerical reliability in simulation, and still in [41], the authors show how sensitive a simulated system is in different processors.

It seems clear that much research has been devoted to investigating the influence of finite precision on digital filters [32,34,36,42,43]. In those investigations, there are many cases where the quality of filter is measured using the filter response or signal-to-noise ratio (SNR) [43]. Despite the fact that the effect of filters on entropy has been pointed out since the work of Shannon [44], there is much less attention given to the entropy effects due to finite precision digital filters on the filtered signal. One work in this direction has been undertaken by Badii et al. [45], who show the influence of an infinite-impulse response in the fractal dimension of the attractor reconstructed from a filtered chaotic signal. Other works have employed entropy to the design of digital filters. For instance, Madan [46] has introduced the use of the maximum entropy method for the design of linear phase FIR digital filters. In [47], another attempt to use entropy in the design of digital FIR filters has been observed. However, no work has been found investigating the effects on entropy on a filtered signal by an IIR filter. This paper seeks to relate the computational limitations and the variation of the main parameters of a filter in the measured entropy. As entropy is a good index to detect increasing of noise in a signal, we have used a boundary technique to observe the effects of finite precision on the parameters of the filters according to the word length of 16 or 32 bits. We noticed that entropy is more sensitive than SNR. It was important to show that despite the ideal linear filter do not increase entropy, numerical experiments using the elliptic, Butterworth and Chebyshev filters have shown an increasing of entropy. Additionally, a positive correlation between order and entropy has been observed in the elliptic filter. This information can be useful to design or to evaluated digital filters in situations where the growth noise should be mitigated.

The remainder of this paper is organized as follows. The definitions of IIR, FIR filters, quantization, and entropy are given in Section 2 as well as three scenarios of the simulation. Section 3 presents the results, where three filter types are investigated: Butterworth, Chebyshev, and elliptic. The remaining section is devoted to summarizing our results.

## 2. Materials and Methods

### 2.1. IIR Filter

IIR digital filters are characterized by having infinite impulse response [48]. They have output feedback, which makes them interesting because they allow achieving a more selective frequency response with lower number of coefficients. IIR digital filters are represented by the following transfer function,
(1)$$H\left(z\right)=\frac{Y(z)}{X(z)}=\frac{\sum_{k=0}^{M}b_{k}z^{-k}}{\sum_{l=0}^{N}a_{l}z^{-l}}$$
(2)$$\sum_{l=0}^{N}a_{l}z^{-l}Y\left(z\right)=\sum_{k=0}^{M}b_{k}z^{-k}X\left(z\right),$$
where *N* and *M* are the degree of the numerator and denominator polynomial, respectively; $b_{k}$ and $a_{l}$ are the filter coefficients. To find the difference equation of the filter, the inverse z-transform of each side of the Equation (Equation 2) is taken. The result is as follows.
(3)$$\sum_{l=0}^{N}a_{l}y\left[n-l\right]=\sum_{k=0}^{M}b_{k}x\left(n-k\right).$$

A more condensed form of the difference equation is
(4)$$y\left[k\right]=\frac{1}{a_{0}}\left(\sum_{k=0}^{M}b_{k}x\left[n-k\right]-\sum_{l=1}^{N}a_{1}y\left[n-l\right]\right).$$
and taking $a_{0}=1$, we have
(5)$$y\left[k\right]=\sum_{k=0}^{M}b_{k}x\left[n-k\right]-\sum_{l=1}^{N}a_{1}y\left[n-l\right].$$

### 2.2. Quantization Error

In the implementation of digital filters, the limitation of finite word length results in coefficient quantization errors, which may have unexpected effect in the frequency response [49]. This quantization error may be seen in a more realistic way if we consider the coefficients of the filter bounded from above and from below. Thus, quantizing can be seen in some way as adding a certain amount of noise. The fewer bits we use in quantization; the more noise is added. This is precisely the noise source shown in Figure 1.

Using a fixed point representation, the quantization error is given by
(6)$$|\epsilon_{r}|\leq \frac{Q}{2},$$
where $Q=2^{b}$ and *b* is the number of bits. Thus, the coefficients of Equation (Equation 5) present lower limits given by
(7)$$\begin{array}{ccc} {\underline{a}}_{k} & = & a_{k}-Q/2 \end{array}$$
(8)$$\begin{array}{ccc} {\underline{b}}_{k} & = & b_{k}-Q/2, \end{array}$$
whereas the upper limits are given by
(9)$$\begin{array}{ccc} {\bar{a}}_{k} & = & a_{k}+Q/2 \end{array}$$
(10)$$\begin{array}{ccc} {\bar{b}}_{k} & = & b_{k}+Q/2. \end{array}$$

This is equivalent to say that the quantization error produces an interval around the desired value of the coefficients. In other words, the approximated value of the coefficients ${\hat{a}}_{k}$ and ${\hat{b}}_{k}$ are given by
(11)$$\begin{array}{ccc} {\underline{a}}_{k}< & {\hat{a}}_{k}< & {\bar{a}}_{k} \end{array}$$
(12)$$\begin{array}{ccc} {\underline{b}}_{k}< & {\hat{b}}_{k}< & {\bar{b}}_{k}. \end{array}$$

### 2.3. Entropy

Entropy reflects a direct relationship between the length of the information and its uncertainty. As entropy quantifies probabilistic and repetitive events, it is utilized so generally in different fields [50]. The maturation of the idea of entropy of random variables and processes by Claude Shannon furnished the origins of information theory. In fact, Shannon’s first name for this concept was *uncertainty* and that was the reason for many to define entropy as “a measure of the uncertainty about the outcome of a random process” [51]. The connection with the digital filter becomes clear when the original scheme proposed by Shannon is noticed. This scheme has been adapted in Figure 1. Shannon was interested in how a message could be transmitted through a channel from a transmitter to a destination. In this process, a key a feature is to consider the presence of noise. Here, we see this scheme from the perspective of filtering. Thus, the channel is our filter, which takes the input and changes it into the output. The noise source in our case comes from the finite precision hardware/software where the digital filter is implemented. It is evident that in real applications many other sources of noise should be considered. Nevertheless, for the purpose of this work, we focus our attention only in the operation of the filter as source of noise.

In Section 22, Shannon [44] states “The operation of the filter is essentially a linear transformation of coordinates.” Shannon deduced this by considering the fact that if an ensemble having an entropy $H_{1}$ per degree of freedom in band *W* is passed through a filter with characteristic Y(f) the output ensemble has an entropy given by Equation (Equation 13). In other words, the new component’s frequencies are just the old ones multiplied by a gain. Moreover, Shannon has described this in such way that a filter presents a direct impact on the entropy of a signal. It is clear from Shannon’s idea that signals filtered by ideal filters high-pass, low-pass, passband, or stopband should have their entropy decreased, as can be seen in [44] (p. 40)
(13)$$H_{2}=H_{1}+\frac{1}{W}\int_{W}log{\left|Y\left(f\right)\right|}^{2}df.$$

There are a few sorts of entropy characterized in the literature. With regards to thermodynamics, entropy alludes to the measure of *disorder*. In statistical mechanics, it refers to the amount of uncertainty in the system. In information theory, it is a proportion of the uncertainty related with a random variable [44,52]. Shannon provides the optimal number of binary digits to represent each event of a given message so that the average number of bits/events of the message is as small as possible. Shannon entropy is defined by [53]
(14)$$H\left(X\right)=\sum_{i=1}^{L}P_{i}log_{2}\frac{1}{P_{i}},$$
where H(X) is the entropy (bits), *X* is a symbol, $P_{i}$ is the probability value of symbol *X*, and *L* is the size of the signal. In our case, we measure the entropy for word lengths of 16 and 32 bits. In a complete random signal represented by a word length of 16 bits, the entropy is exactly 16 bits.

To proceed with the calculation of Shannon entropy, we apply the following standardization process to the output signal as follows,
(15)$$S_{k}=\mathrm{ceil}\left(\frac{y_{k}-\min\left(y_{k}\right)}{\max(y_{k}-\min\left(y_{k}\right))}\times 2^{WL}\right),$$
where $y_{k}$ is the signal; ceil(x) is a function that returns the smallest integer not less than x; min and max return the lowest and the largest value from a vector, respectively; and WL is the word length given in bits. Figure 2a,b presents a sinusoidal wave
y(k)=2sin(2π2)t
sampled at δ=0.01 to illustrate this procedure. For this sine the calculated entropy using WL=8 and Equation (Equation 14) is H=4.71. A uniform distributed random signal is shown in Figure 2c, for which the calculated entropy is H(k)=7.59±0.03. Increasing the number of samples of the random signal, the entropy value approaches 8, as expected.

A last observation regarding this procedure is related to the need to discard the transient and limit the number of samples for calculating the entropy of filtered signals. The number of samples has been adopted as $2^{10}$, which limits the measure of the entropy up to 10. Only in one table, we have adopted $2^{12}$ samples. Tests made with greater number of samples showed us that this limit is sufficient to a reliable estimation of Shannon entropy in this work.

### 2.4. Entropy to Detect Noise

Entropy has been widely used to detect noise in signal and images [52,54,55,56]. To show the effectiveness of entropy as a way to detect growth of noise in a signal, we have calculated the entropy changing the variance in Gaussian noise from σ=0.01 to σ=0.02. The mean has been kept as μ=0. A sine wave is shown in Figure 3a. Gaussian noise with σ=0.01 and σ=0.02 has been added to this sine wave and shown in Figure 3b,c, respectively. The calculated entropy are (a) 5.66, (b) 5.95±0.03, and (c) 6.23±0.04. The level of Gaussian noise is quite unseeingly, yet the entropy has been sensitive for the increasing of noise.

Entropy is a sensitive way to measure uncertainty. To further show this property, let us compare this measure with the well-known signal-to-noise ratio (SNR) given in dB by the following equation
(16)$$SNR=20\log_{10}\frac{A_{\mathrm{signal}}}{A_{\mathrm{noise}}}$$
where *A* is root mean square (RMS) amplitude. Let the relation between the entropy of signal and noise (ESN) be
(17)$$ESN=20\log_{10}\frac{H_{\mathrm{signal}}}{H_{\mathrm{noise}}},$$
where *H* is the entropy of the signal and noise. Using these two equations, we are going to compare the sensitivity in a little variation of noise. Table 1 shows the difference between SNR and ESN for the signal of Figure 1b (sine wave with Gaussian noise of σ=0.01) and the same signal but with a σ given in the first column of Table 1. The message of this table is simple. For the case of σ=0.0200, the SNR gives a difference of 2.6359±0.6920 dB, whereas the entropy for this difference is 15.9343±3.3038 dB. When the difference between the variance of the noise of these two signals are only 0.0125−0.01=0.0025, we have more confidence to use the ESN to detect this level of difference of noise, as the difference between the SNR of these two signal is 0.527±0.589, whereas for ENS we have 4.023±2.866. For the SNR case, the interval given by one σ is (−0.062; 1.116) and we have lost the confidence to ensure that one of the signals presents a higher level of noise than other.

### 2.5. Numerical Experiments

In this section, three numerical experiments are described. For each experiment, the main steps are outlined. All the numerical experiments have been performed in Octave [57] on a Windows computer. These routines are available upon request. These experiments have been designed to check some effects of finite precision in entropy of digital filtered signals. In the Numerical Experiment 1, poles and zeros are perturbed by quantization error due to a 16- and 32-bit fixed point representation. Numerical Experiment 2 aims at examining the increasing of entropy using the elliptic filter. The correlation between order and entropy increasing is verified in the Numerical Experiment 3.

#### 2.5.1. Numerical Experiment 1

The proposed scheme can be summarized in the following steps.

 **Step** **1:** Use the commands butter, cheby, or ellip of Octave to generate the poles and zeros of the transfer function according to Equation (Equation 1). **Step** **2:** Choose number of bits and calculate quantization error according to Equation (Equation 6). **Step** **3:** Insert the quantization error at the poles and zeros. Using a similar strategy adopted in [49], Equation (Equation 5) can be rewritten as follows.
(18)$$y_{1}\left[k\right]=\sum_{k=0}^{M}\underline{b_{k}}x\left[n-k\right]-\sum_{l=1}^{N}{\underline{a}}_{l}y\left[n-l\right].$$ **Step** **4:** The signal is filtered using 50 different combinations described by Equations (Equation 7) and (8). **Step** **5:** Apply the standardization procedure to the filtered signal according Equation (Equation 15). **Step** **6:** Calculate the mean and standard deviation of the entropy from the 50 filtered signals.

In Numerical Experiment 1, the filter poles and zeros are perturbed with the effects of 16- and 32-bit quantization. The input signal is composed as a sum of sinusoidal signals of 50, 75, 125, and 150 Hz. The order of the filters is given in Table 2.

#### 2.5.2. Numerical Experiment 2

The following steps outline the Numerical Experiment 2.

 **Step** **1:** Use the command ellip of Octave to generate the poles and zeros of the transfer function (Equation (Equation 1)). **Step** **2:** Choice of input signal (Table 3). **Step** **3:** The signal is filtered using 50 values of WL within 1024 to 6024. **Step** **4:** Apply the standardisation procedure to the filtered signal according Equation (Equation 15) **Step** **5:** Filter signal using Equation (Equation 5). **Step** **6:** Compute the mean and standard deviation of entropy of the filtered signal.

In Numerical Experiment 2, entropy was calculated for the original signal in Table 3 and the filtered signal using elliptic filters. To compare, the input signal was simulated without the filtered frequency components. The complete description of the input signal and the ideally filtered signal can be seen in Table 3. The variation of the length of the signal has been used here to calculate mean and standard deviation of the entropy.

#### 2.5.3. Numerical Experiment 3

The following steps describe the Numerical Experiment 3.

 **Step** **1:** Use the command butter, cheby, or ellip of Octave to generate the poles and zeros of the transfer function, Equation (Equation 1). **Step** **2:** Filter signal using Equations (Equation 5). **Step** **3:** The signal is filtered using 50 different values of WL within 1024 to 6024. **Step** **4:** Apply the standardisation procedure to the filtered signal according Equation (Equation 15). **Step** **5:** Compute the mean and standard deviation of entropy of the filtered signal. **Step** **6:** Change the order of the filter from 1 to 8 for each of the steps 1 to 5.

In this experiment, the filter order was varied for an input signal with frequencies of 20, 60, and 80 Hz and the cut-off frequency of 60 Hz.

## 3. Results

The results of Numerical Experiment 1 are shown in Table 4. Table 5 shows the result of the Numerical Experiment 2, whereas Table 6 and Figure 4 show the results of Numerical Experiment 3.

## 4. Discussion and Conclusions

This work has investigated the effects of finite precision in the entropy of digital filtered signals. This allows us to quantify the introduction of noise due the action of such filters. We have shown that entropy is a good alternative to identify the presence of noise. It has presented a better result than the signal-to-noise ratio for small amount of variance. To observe the effects of entropy in filtered signals, we have designed three numerical experiments. In Numerical Experiment 1, we have evidenced the increasing of the entropy in all types of filters investigated (Butterworth, Chebyshev, and elliptic) for 16 and 32 bits. The entropy of the input signal is H=4.9255, whereas in all the filtered signal the entropy is H>5.32. This is not what is expected for an ideal linear filter (see [44]). We should notice, according to Table 2, that elliptic has been set up with the lowest order. Even in such circumstances, this type of filter has shown practically the same level of entropy in the filtered signal.

The results of Numerical Experiment 2 are shown in Table 5. In this case, an ideally filter is simulated by taking out some of the frequency components of the signal. The entropy of filtered signal has been significantly increased varying from 6.5 to almost 8.

In Numerical Experiment 3, we have noticed another feature as described in Table 6. This experiment shows a significant positive correlation at the 0.05 level (2-tailed) for elliptic with *p*-value equals to 0.030. From these experiments, it seems clear that the elliptic filter introduces more uncertainty, that is, entropy, to the filtered signal when compared to Butterworth and Chebyshev filters. Figure 4 shows the FFT of the signals. It is possible to notice a slight difference between subfigures (b) and (c).

The remarks made in this manuscript is coherent to what have been presented by DeBrunner et al. [47]. As we are focusing our attention to the source noise furnished by arithmetical operations (see Figure 1), design strategies that look for more efficient ways to implement mathematical expressions can be useful to reduce entropy. In our future work, we intend to test different topologies of filter (direct or cascade, for instance) to verify its influence in the increasing of entropy as done in this manuscript. This seems a quite reasonable pathway as the order is related to the increasing of the number of mathematical operations, which is a well-known source of the noise. We also intend to investigate the influence of sample rate and the number of samples in the computation of entropy.

## Figures and Tables

**Figure 1 entropy-22-00365-f001:**
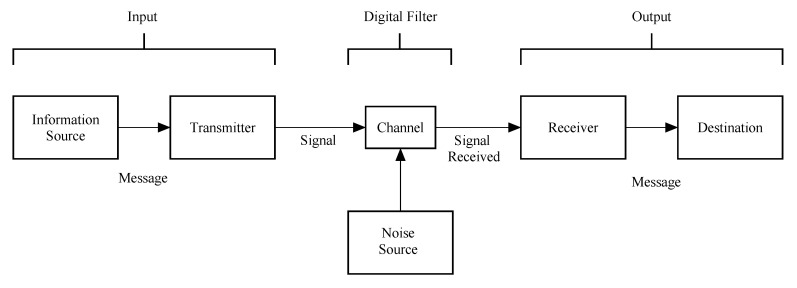
Adaptation of “Schematic diagram of a general communication system” in [44]. In our case, we are interested in looking the channel as a filter and noise source as a consequence of finite precision implementation of the digital filters.

**Figure 2 entropy-22-00365-f002:**
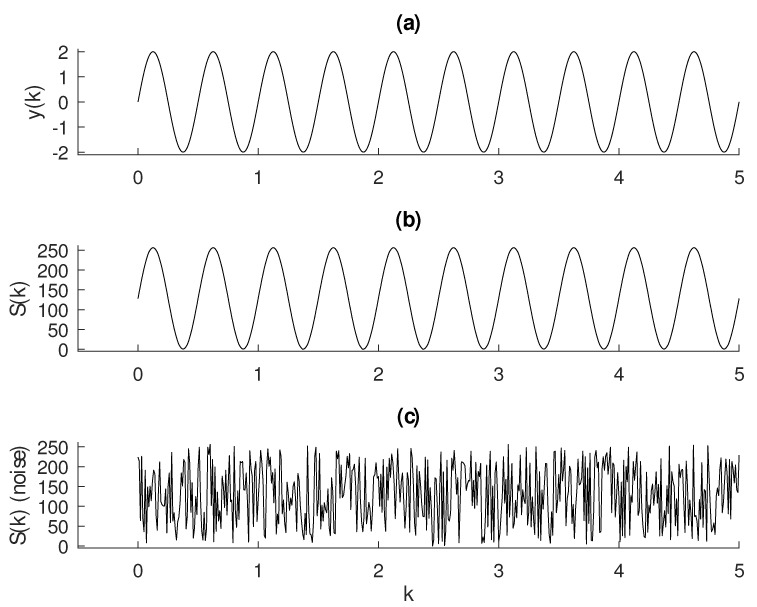
Procedure of standardization of the vector y(k) in panel (**a**) for entropy calculation using Equation (Equation 15). The result S(k) in panel (**b**) is composed of only integers within 0 to $2^{WL}-1$. In this case, we have used WL=8 bits. Panel (**c**) shows a random signal with uniform distribution. Fifty runs of this signal produces an entropy of H=7.59±0.03. Increasing the number of samples, this value approaches 8, as expected.

**Figure 3 entropy-22-00365-f003:**
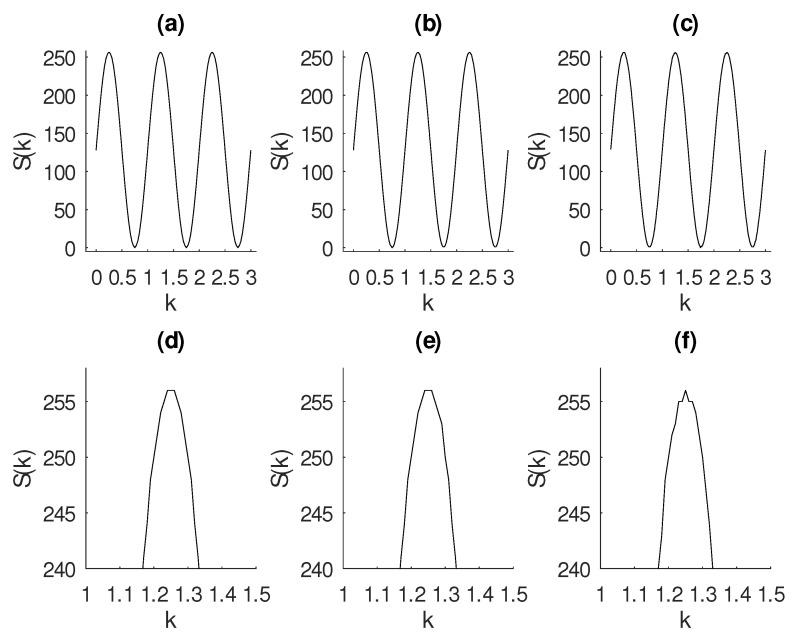
Computation of Shannon entropy for three signals. All signals have been standardized according procedure described in Equation (Equation 15). (**a**) Sine wave 2sin(2π2t). (**b**) Sine wave added with Gaussian noise of σ=0.01 and μ=0. (**c**) Sine wave added Gaussian noise of σ=0.02 and μ=0. The calculated entropy are (**a**) 5.66, (**b**) 5.95±0.03, and (**c**) 6.23±0.04. The level of Gaussian noise is quite unseeingly; yet the entropy has been sensitive for the increasing of noise. Panels (**d**–**f**) show a zoom in the above figure to see the presence of a small level of noise in the signals of panels (**b**,**c**).

**Figure 4 entropy-22-00365-f004:**
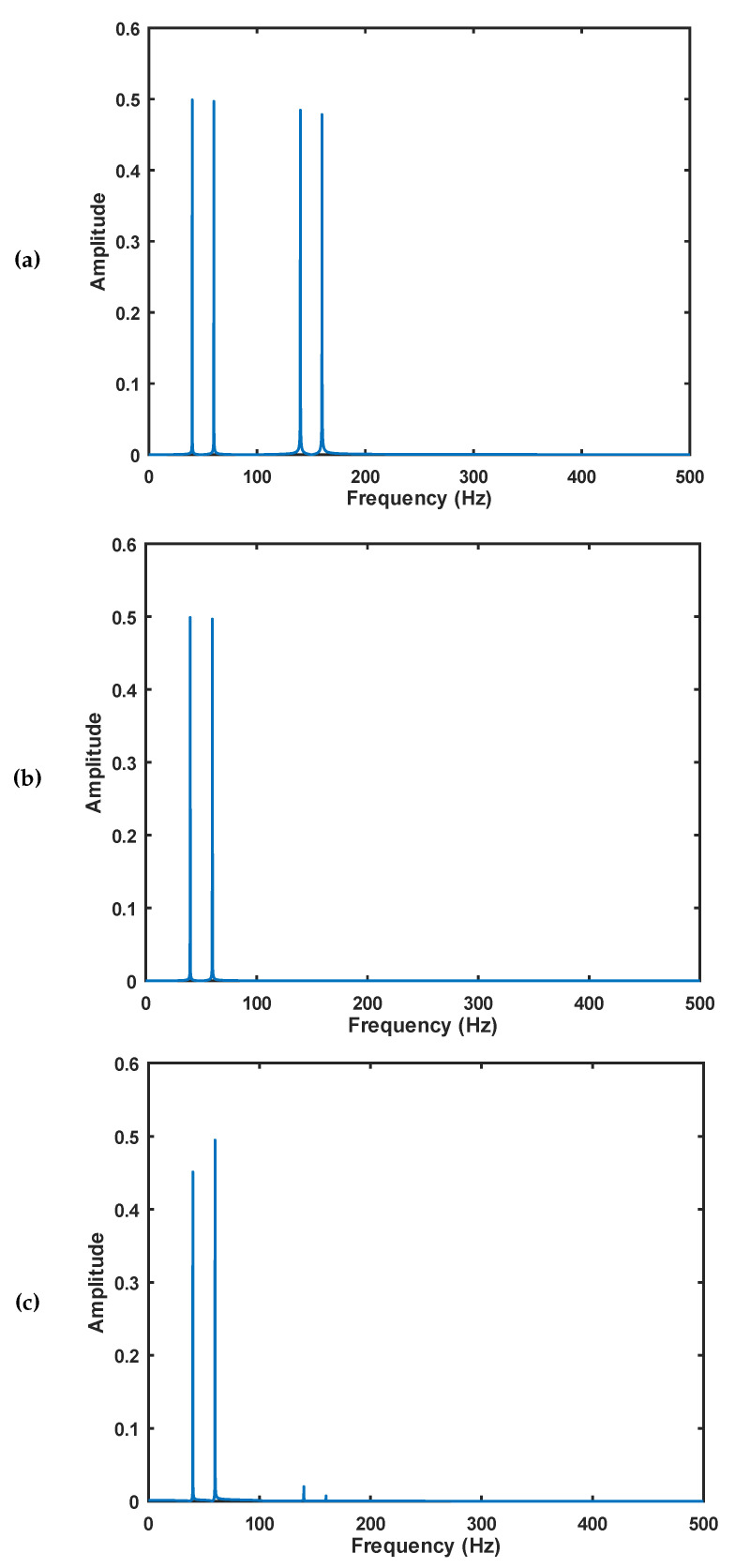
(**a**) FFT of signal 2, Table 3—Complete signal; (**b**) FFT of signal 2 in Table 3—Ideally Filtered signal; (**c**) FFT of Chebyshev filter. The FFT computations the expected similarity between the signals. This is another point that makes relevant to investigate the effect of digital filter in the entropy of the filtered signal.

**Table 1 entropy-22-00365-t001:** Sensitivity to variation of the noise for SNR and ESN. The reference signal has been added with Gaussian noise of σ=0.01. Here, we show the difference between the measure of SNR and ESN in relation of this signal. Observe that an increasing of 0.01 in the standard deviation of the signal is able to increase 15.934 in the ESN (entropy) and only 2.635 for the signal-to-noise ratio (SNR). Fifty runs have been employed to calculate mean and standard deviation of these quantities in dB. When the difference between these two signals is only 0.002, the ESN would be more robust to detect this difference than the SNR.

σ	Difference of SNR (dB)	Difference of ESN (dB)
0.0200	2.635±0.692	15.934±3.303
0.0192	2.571±0.664	14.928±2.710
0.0183	2.104±0.673	14.110±2.858
0.0175	1.905±0.538	11.525±2.835
0.0167	1.648±0.582	11.481±3.356
0.0158	1.509±0.728	9.620±3.438
0.0150	1.265±0.741	8.498±4.123
0.0142	0.870±0.650	6.720±3.219
0.0133	0.745±0.623	4.817±3.229
0.0125	0.527±0.589	4.023±2.866

**Table 2 entropy-22-00365-t002:** Order of the filters for Numerical Experiment 1. We have adopted 100 Hz as cut-off frequency in the case of low and high pass. Passband and stopband filters have been designed with 70 and 130 Hz.

Type Filters	Butterworth	Chebyshev	Elliptic
High-pass	14	8	6
Low-pass	14	8	6
Passband	5	4	3
Stopband	5	4	3

**Table 3 entropy-22-00365-t003:** Input signals for the Numerical Experiment 2. We have designed three types of signals composed by different summation of harmonics. The values of frequencies 1–6 are 40 Hz, 60 Hz, 80 Hz, 130 Hz, 150 Hz, and 170 Hz, respectively. To compare, the input signal was simulated without the filtered frequency components as shown in the third column. This is equivalent to produce an output by an ideal filter. In all cases, a sample rate of 0.001 s has been adopted. Different values or even variable sample rate has not been investigated in this work and let for future research.

Signal	Complete Signal	Ideally Filtered Signal
1	$sin\left(w_{1}t\right)+sin\left(w_{6}t\right)$	$sin(w_{1}t)$
2	$sin\left(w_{1}t\right)+sin\left(w_{2}\right)+sin(w_{5}t+sin\left(w_{6}t\right)$	$sin\left(w_{1}t\right)+sin\left(w_{2}\right)$
3	$sin\left(w_{1}t\right)+sin\left(w_{2}\right)+sin\left(w_{3}\right)+sin\left(w_{4}\right)+sin\left(w_{5}\right)+sin\left(w_{6}t\right)$	$sin\left(w_{1}t\right)+sin\left(w_{2}\right)+sin\left(w_{3}\right)$

**Table 4 entropy-22-00365-t004:** Results of the Numerical Experiment 1. Entropy calculation for Butterworth, Chebyshev, and elliptic filter. We have applied our test in 50 types of filters in two word length (WL): 16 and 32 bits. The mean μ and standard deviation σ of the 50 results are shown. Values of σ shown as 0.0000 means that calculated values are lower than 0.00005. The measure of the entropy for the original signal is H=4.9255. From this result, it is clear the increasing in the measured entropy in all filtered signals.

Type Filters	WL	Butterworth	Chebyshev	Elliptic
Low-pass	16	5.3276±0.0000	5.3256±0.0099	5.3775±0.1639
	32	5.5451±0.0487	5.3529±0.0176	5.3297±0.0055
High-pass	16	5.3266±0.0070	5.3266±0.0072	5.3290±0.0068
	32	5.7350±0.0594	5.3287±0.0060	5.3297±0.0058
Passband	16	5.3276±0.0000	5.3266±0.0070	5.3276±0.0000
	32	5.3726±0.0302	5.3371±0.0156	5.3284±0.0031
Stopband	16	5.3266±0.0072	5.3276±0.0000	5.3266±0.0070
	32	5.4125±0.0410	5.3314±0.0079	5.3287±0.0048

**Table 5 entropy-22-00365-t005:** Entropy of the original signal, the simulated signal without the frequency components and the filtered signal. The third column is an ideal filtering. As expected, the entropy is reduced. The same does not occur with the use of designed filter based on elliptic type. These tests have been used 32 bits. Similar results have been obtained for Butterworth and Chebyshev. Mean and standard deviation have been calculated over 50 runs for lengths from 1024 to 6024 samples of the signal.

Signal	Original	Ideally Filtered	Elliptic
1	6.5658±0.0006	6.5435±0.0004	7.9828±0.0240
2	6.5131±0.0012	6.4839±0.0009	7.9586±0.0235
3	6.5654±0.0008	6.5434±0.0004	7.9568±0.0239

**Table 6 entropy-22-00365-t006:** Entropy calculation with order variation for three types of filters. We have used the software PSPP to perform the regression and significance analysis. A significant positive correlation between order and filter types are found only for elliptic filter. Although there is r=0.7 for Chebyshev, its *p*-value = 0.052 which means that there is no statistical significance. Correlation is significant at the 0.05 level (2-tailed) for elliptic with *p*-value equals to 0.030. We have performed our calculations using word length of 32 bits. The values are average of 50 runs for different length of time series within 1024 to 6024 samples.

Order	Butterworth	Chebyshev	Elliptic
1	6.6458±0.0006	6.6458±0.0006	6.6458±0.0006
2	6.6458±0.0006	6.6458±0.0006	6.6458±0.0006
3	6.6458±0.0006	6.6470±0.0009	6.6458±0.0006
4	6.6458±0.0006	6.6458±0.0006	6.6458±0.0006
5	6.6458±0.0006	6.6458±0.0006	6.6458±0.0006
6	6.6458±0.0006	6.6458±0.0006	6.6508±0.0026
7	6.6458±0.0006	6.6533±0.0012	11.6121±0.7048
8	6.6458±0.0006	6.6683±0.0056	11.6121±0.7048
*r*	–	0.700	0.760
*p*-value	–	0.052	0.030

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
