# Peer review of "Some Remarks about Entropy of Digital Filtered Signals"

_entropy, 2020, doi:10.3390/e22030365_

Round 1
Reviewer 1 Report
File

Author Response
Please see file attached.

Reviewer 2 Report
The paper presents some numerical concerning the entropy of digital filtered signals. The presented results seem simple and correct.
My main suggestion are:
1. The authors should clearly explain in the beginning of the text why a higher or lower entropy in the filtered signal matters. Why this knowledge is useful in projecting a digital filter?
2. The results in Tables 3-6 are deterministic? What is the precision in the results? Standard deviations should be presented together with the main value (XX ± XX) so that one can compare the entropy diferences between word lengths.
As minor remarks:
- There are many typos and bad-written sentences along the text. It must be thoroughly revised.
- There are typos in equations. See for instance, eq. (12) and Table 2.
Author Response
Please report to attached file.

Round 2
Reviewer 1 Report
The authors of the manuscript entitled “Some remarks about entropy of digital filtered signals” succeeded to shiftily address all the suggestions that I had in my first round of the review processed.
The manuscript details now properly the numerical experiments and explains the obtained results.
The resulted paper is valuable and makes a real service to authors' research work.